# Ratanjot (*Alkanna tinctoria* L.) Root Extract, Rich in Antioxidants, Exhibits Strong Antimicrobial Activity against Foodborne Pathogens and Is a Potential Food Preservative

**DOI:** 10.3390/foods13142254

**Published:** 2024-07-17

**Authors:** Annada Das, Subhasish Biswas, Kaushik Satyaprakash, Dipanwita Bhattacharya, Pramod Kumar Nanda, Gopal Patra, Sushmita Moirangthem, Santanu Nath, Pubali Dhar, Arun K. Verma, Olipriya Biswas, Nicole Irizarry Tardi, Arun K. Bhunia, Arun K. Das

**Affiliations:** 1Department of Livestock Products Technology, West Bengal University of Animal and Fishery Sciences, Kolkata 700037, India; dasannada.555@gmail.com (A.D.); lptsubhasish@gmail.com (S.B.); lptgopal@gmail.com (G.P.); sushmitamoirangthem@gmail.com (S.M.); 2Department of Veterinary Public Health and Epidemiology, Faculty of Veterinary and Animal Sciences, Banaras Hindu University, Mirzapur 231001, India; kaushik_vph@bhu.ac.in; 3Department of Livestock Products Technology, Faculty of Veterinary and Animal Sciences, Banaras Hindu University, Mirzapur 231001, India; dr.dipanwita.vet@gmail.com; 4Eastern Regional Station, ICAR-Indian Veterinary Research Institute, 37 Belgachia Road, Kolkata 700037, India; npk700@gmail.com (P.K.N.); santanu.vet03@gmail.com (S.N.); 5Laboratory of Food Science and Technology, Food and Nutrition Division, University of Calcutta, 20B, Judges Court Road, Alipore, Kolkata 700027, India; pubalighoshdhar@yahoo.co.in; 6Goat Products Technology Laboratory, ICAR-Central Institute for Research on Goats, Makhdoom, Mathura 281122, India; arun.lpt2003@gmail.com; 7Department of Fishery Engineering, West Bengal University of Animal and Fishery Sciences, Kolkata 700037, India; olipriya.online16@gmail.com; 8Molecular Food Microbiology Laboratory, Department of Food Science, College of Agriculture, Purdue University, West Lafayette, IN 47907, USA; irizarrn@purdue.edu; 9Department of Comparative Pathobiology, College of Veterinary Medicine, Purdue University, West Lafayette, IN 47907, USA

**Keywords:** *Alkanna tinctoria* L., Ratanjot, antimicrobial, antioxidant, toxicity, probiotics, sensory attributes, food preservation, food safety, foodborne pathogens

## Abstract

Natural and sustainable plant-based antioxidants and antimicrobials are highly desirable for improving food quality and safety. The present investigation assessed the antimicrobial and antioxidant properties of active components from *Alkanna tinctoria* L. (herb) roots, also known as Ratanjot root. Two methods were used to extract active components: microwave-assisted hot water (MAHW) and ethanolic extraction. MAHW extract yielded 6.29%, while the ethanol extract yielded 18.27%, suggesting superior Ratanjot root extract powder (RRP) solubility in ethanol over water. The ethanol extract showed significantly higher antioxidant activity than the MAHW extract. Gas Chromatography–Mass Spectrometry analysis revealed three major phenolic compounds: butanoic acid, 3-hydroxy-3-methyl-; arnebin 7, and diisooctyl pthalate. The color attributes (L*, a*, b*, H°ab, C*_ab_) for the ethanolic and MAHW extracts revealed significant differences (*p* < 0.05) in all the above parameters for both types of extracts, except for yellowness (b*) and chroma (C*_ab_) values. The ethanol extract exhibited antimicrobial activity against 14 foodborne bacteria, with a significantly higher inhibitory effect against Gram-positive bacteria (*Listeria monocytogenes* and *Staphylococcus aureus*) than the Gram-negative bacteria (*Salmonella enterica* serovar Typhimurium and *Escherichia coli*). The minimum inhibitory concentration (MIC) and minimum bactericidal concentration (MBC) were both 25 mg/mL for the Gram-negative bacteria, while the MIC and MBC concentrations varied for Gram-positive bacteria (0.049–0.098 mg/mL and 0.098–0.195 mg/mL) and the antimicrobial effect was bactericidal. The antimicrobial activities of RRP extract remained stable under broad temperature (37–100 °C) and pH (2–6) conditions, as well as during refrigerated storage for 30 days. Application of RRP at 1% (10 mg/g) and 2.5% (25 mg/g) levels in a cooked chicken meatball model system prevented lipid oxidation and improved sensory attributes and retarded microbial growth during refrigerated (4 °C) storage for 20 days. Furthermore, the RRP extract was non-toxic when tested with sheep erythrocytes and did not inhibit the growth of probiotics, *Lacticaseibacillus casei*, and *Lactiplantibacillus plantarum*. In conclusion, the study suggests that RRP possesses excellent antimicrobial and antioxidant activities, thus making it suitable for food preservation.

## 1. Introduction

Synthetic antioxidants and antimicrobials have historically been employed to mitigate lipid oxidation, food spoilage, and foodborne infections. However, the ever-increasing consumer concerns regarding food safety [1] coupled with the potential toxicity [2] of these synthetic or chemical compounds, like butylated hydroxyanisole (BHA), butylated hydroxytoluene (BHT), propyl gallate (PG), and nitrites, have spurred a quest within the food industry for natural, sustainable alternatives [3,4,5]. Similarly, the emergence of antimicrobial resistance (AMR) in medicine has increased attention to discovering new plant, animal, or microbial antimicrobial compounds [6,7]. Natural products, especially plant components, are composed of a vast array of bioactive compounds, like polyphenols, flavonoids, alkaloids, tannins, ketones, aldehydes, alcohols, and esters, with antioxidant and antimicrobial compounds [8,9],and possess non-toxic, non-antigenic, and biocompatible properties, hence are utilized in various industries [10,11]. In the food industry, plant extracts and essential oils are gaining popularity, as the majority of plants used in traditional food and medicines are safe as perceived by consumers and received the GRAS (Generally Recognized As Safe) status by the United States Food and Drug Administration (FDA) due to their long history of use [12,13,14]. Further, around 75% of the world’s rural populace relies on plant materials included in traditional systems of medicine [15].

*Alkanna tinctoria* L., commonly known as dyer’s alkanet, alkanet, dyer’s bugloss, orchanet, or Ratanjot (in India), of the *Boraginaceae* family, is a perennial herb native to Europe and the Mediterranean region and Asia, including the Middle East [16,17,18]. The plant is characterized by its striking blue flowers and dark red roots [19,20] (Figure 1). The plant is most renowned for the vibrant red dye extracted from its roots, which has been treasured for centuries across various cultures and industries [16,21]. The natural red pigments in Ratanjot roots have a broader application in medicine, and the food, cosmetic, and textile industries [22,23].

In Ayurveda, the traditional Indian medicine, Ratanjot root is used to treat various ailments, including skin disease, digestive disorders, urinary calculi, menstruation abnormalities, and respiratory illness [16,24]. The Ratanjot root contains bioactive components known for their anti-inflammatory, antimicrobial, and wound-healing properties [25,26,27]. Moreover, Ratanjot root or alkanet extracts have shown anticancer activities [27,28] and are also used in histological staining of tissue samples [18,21]. However, no scientific report on the toxicity profile of Ratanjot roots is found in the literature.

In the food industry, Ratanjot (*A. tinctoria* L.) root is mainly used as a natural food coloring and flavoring agent by virtue of its two important naphthoquinone compounds, i.e., alkannins and shikonins, which are responsible for color [24] and the phenolic compounds which contribute to flavor [17]. In Indian culinary practices, Ratanjot roots are used to give the captivating reddish hue and subtle earthy flavor to signature Kashmiri meat dishes, like *Rogan josh* [24,29]. It is also used in sweets, beverages, and savory dishes to improve sensory attributes, such as color and flavor [30].

However, very few studies have been conducted to understand theantioxidant, antimicrobial, and cytotoxicity effects of *A. tinctoria* L. leaves and root extracts [31,32]. The scarcity of research proving the antimicrobial and antioxidant properties of *A. tinctoria* L. roots might have arisen from limited research focus, overshadowed by its primary use as a dye in the textile industry. Moreover, there are no available scientific reports on the application of *Alkanna tinctoria* L. roots in the cooked meat-model system. The present study aimed to unveil and validate thein vitro antimicrobial and antioxidant activities of *A. tinctoria* L. or Ratanjot roots in a meat-model system for potential food industry application.

## 2. Materials and Methods

### 2.1. Chemicals and Media

Chemicals and analytical and molecular biology grade reagents were purchased from Sigma-Aldrich (Saint Louis, MO, USA) and Sisco Research Laboratories (SRL) (Mumbai, India). Media for bacterial culture were purchased from Hi-Media and SRL (Mumbai, India).

### 2.2. Bacterial Cultures and Growth Conditions

*Staphylococcus aureus* ATCC 25923, *Listeria monocytogenes* ATCC 19111, *Listeria monocytogenes* ATCC 13932, *Salmonella enterica* serovar Typhimurium ATCC 14028, *Escherichia coli* ATCC 25922, *Lactiplantibacillus plantarum* MTCC 2621 and *Lacticaseibacillus casei* MTCC 1423 and nine field isolates of *Staphylococcus aureus* isolated from poultry, goat and swine meat (Table 1) were obtained from the Department of Veterinary Public Health and Epidemiology, Banaras Hindu University, India. The foodborne pathogenic bacterial cultures and the lactobacilli were grown in brain heart infusion (BHI) broth and De Man–Rogosa-Sharpe (MRS) broth, respectively at 37 °C for 18–24 h and 48 h under aerobic conditions, before being used in the study.

### 2.3. Preparation of Ethanol and Microwave-Assisted Hot Water Extracts of Ratanjot (Alkanna tinctoria L.) Root and Characterization

The dried Ratanjot roots (Figure 1) collected from the local market were identified by a botanist, an ex-professor of Samanta Chandra Sekhar College (Autonomous), Puri, affiliated to Utkal University, Odisha, India. The Ratanjot roots were further dried in a food drier (Ambay Biotech, Delhi, India) at 40 °C overnight, chopped into small pieces, pulverized (500 W Mixer Grinder, Philips India Limited, Delhi, India), and sieved (using 60 mesh) to obtain a fine powder, designatedas Ratanjot root powder (RRP). The RRP was stored in a sterile, amber-colored glass bottle at room temperature (25 ± 1 °C) until use. The microwave-assisted hot water (MAHW) and ethanol extracts (EE) of RPP were prepared as described before [32,33] with slight modifications.

For the ethanol extraction method, 20 g RRP was extracted in 100 mL of 70% ethanol (*v*/*v*) by continuously shaking in a shaker incubator (Remi, Mumbai, India) for 24 h at 37 °C. In the MAHW extraction method, 20 g of RRP was extracted with 100 mL of sterile distilled water in a mono-mode microwave oven (NEOS-GR, Milestone SRL, ViaFatebenefratelli, Sarisole BG, Italy) equipped with a closed vessel system. The power setting was adjusted to 300 W and the temperature rose to 95 °C within 10 min, extracting the water-soluble components. The samples were then cooled to room temperature. The entire content was centrifuged at 5000× *g* for 10 min (Remi, India), followed by filtration through Whatman filter paper (No. 1). In both methods, the extraction processes were repeated with the residue obtained, and the filtrates from both batches were concentrated using a rotary evaporator (HS-2005 VN, Hahnshin S&T Co., Ltd., Gimpo-si, Gyeonggi-do, Republic of Korea) at 40 ± 1 °C overnight under reduced pressure.

The concentrated extracts were air-dried at room temperature and weighed to calculate the total extractable components (TEC). The dried crude extracts were diluted with sterile distilled water to obtain a concentration of 200 mg/mL and used as the stock preparation for antimicrobial and antioxidant activity assays. The TEC was calculated using the formula: TEC% = (Weight of extracted component/Weight of RRP) × 100.

The pH values of the RRP extracts were recorded using a digital pH meter (pH Tutor, Eutech Instruments, Singapore). The color of the RRP extracts was measured using a ColorQuest XE colorimeter (HunterLab, Reston, VA, USA). In brief, about 20 mL of each extract was placed in a 20 mm cell. The CIELAB parameters, such as lightness (*L**), redness (*a**), yellowness (*b**), hue angle (H°_ab_) and chroma (C*_ab_), were recorded in total transmittance mode, standard illuminant D65 and observer angle 10 degrees as per the manufacturer’s instruction manual (HunterLab, Reston, VA, USA).

### 2.4. Antioxidant Activity Analysis

#### 2.4.1. Total Phenolic Content

The total phenolic content (TPC) of the RRP extract was estimated by the classical Folin–Ciocalteu (FC) method [34] with minor modifications. Briefly, 0.75 mL of 0.2 N FC reagent was added to 100 µL of diluted RRP extract (20 mg/mL), prepared by diluting the stock extract ten times with deionized distilled water. After mixing, the reaction was carried out for 5 min at room temperature. After that, 0.4 mL of 7.5% *w*/*v* Na_2_CO_3_ solution was added, mixed gently, and incubated at room temperature in the dark for 30 min to observe visible deep blue color development. Absorbance was measured at 760 nm against a solvent blank using a UV-visible spectrophotometer (Spectra Max^®^ M5, Molecular Devices, San Jose, CA, USA), and gallic acid at different concentrations was used as the standard. The TPC of the RRP extract was calculated from the standard curve plotted using different concentrations of gallic acid and expressed as gallic acid equivalent (GAE) in mg/g of RRP on a dry weight basis.

#### 2.4.2. 2,2-Diphenyl-1-Picrylhydrazyl (DPPH) Radical Scavenging Assay

The DPPH radical scavenging activity (RSA%) of RRP extract was assayed as before [35] with slight modification. A solution of 0.1 mM (*w*/*v*) DPPH was prepared by mixing 3.94 mg of DPPH in 100 mL methanol. About 1 mL of 0.1 mM DPPH methanol solution was added to an equal volume (1 mL) of the RRP extract at different concentrations (1–20 mg/mL), mixed well, and incubated in the dark at room temperature for 30 min. The color change (from deep violet to light yellow) was measured spectrophotometrically at 517 nm against a solvent blank (DPPH methanolic solution without sample). BHT and BHA at a concentration range 0.01–0.1% (*w*/*v*) were used as positive controls or standards. The DPPH RSA (%) was calculated using the following formula:DPPH RSA (%) = ((A_B_ − A_S_)/A_B_) × 100
where A_B_ = Absorbance of blank and A_S_ = Absorbance of the reaction mixture with the sample. The half maximal inhibitory concentration (IC_50_) value was calculated by plotting a graph using the RSA (%) against different sample concentrations.

#### 2.4.3. 2,2-Azino-Bis (3-Ethylbenzothiazoline-6-Sulfonic Acid) (ABTS^.+^) Radical Cation Decolorization Assay

The ABTS^.+^ radical scavenging activity was determined as before [36] with slight modifications. The ABTS^.+^ radical stock solution was prepared using an equal volume (1:1 *v*/*v*) of 7 mM ABTS^.+^ in water and 2.45 mM potassium persulfate, stored in the dark at room temperature for 16 h. The ABTS^.+^ radical stock solution was then diluted adequately with absolute ethanol to obtain an absorbance of 0.700 ± 0.05 at 734 nm. A 0.1 mL of the RRP extract at different concentrations was added to 3.9 mL of the diluted ABTS^.+^ radical solution, and the color change (from turquoise blue to colorless) was measured after 6 min at 734 nm using a UV-visible spectrophotometer (Spectra Max^®^ M5, USA). The blank was prepared with the diluted ABTS^.+^ radical solution, whereas BHT and BHA at a concentration range 0.01–0.1% (*w*/*v*) were used as standards. The ABTS^.+^ RSA(%) was calculated using the following formula:ABTS^.+^ RSA(%) = ((A_B_ − A_S_)/A_B_) × 100,
where A_B_ = Absorbance of blank and A_S_= Absorbance of the reaction mixture with the sample. The IC_50_ value was calculated by plotting a graph using the ABTS^.+^ RSA (%) against different sample concentrations.

#### 2.4.4. Identification of Phytochemicals Using Gas Chromatography–Mass Spectrometry (GC–MS/MS)

For identification of phytochemicals, especially the phenolic compounds present in RRP extract, GC–MS/MS analysis was carried out using a gas chromatograph mass spectrometer (Shimadzu GCMSQP2010, Nakagyoku, Kyoto, Japan) interfaced with a turbo quadrupole mass spectrometer, fitted with an Rtx-5 fused silica capillary column with dimensions of 30 m in length, 0.25 mm internal diameter and thickness of 0.25 µm, coated with a non-polar stationary phase composed of 5% diphenyl and 95% dimethylpolysiloxane. Methanol was used as the solvent to make dilutions, the total run time was ~70 min, and GC–MS solution software (Version 4.52, Shimadzu Sci, Nakagyoku, Kyoto, Japan)was used for the data analysis [37]. The phytochemical compounds in the RRP extract were identified by matching their mass spectra with standards from the National Institute of Standards and Technologies (NIST Version 20, Gaithersburg, MD, USA). The RRP extract samples were analyzed in triplicate by GC–MS/MS.

### 2.5. Antimicrobial Activity Assays

#### 2.5.1. Agar Well Diffusion Assay

The hot water and ethanol extracts of RRP were filter-sterilized by passing through syringe filters (0.45 µm) and were evaluated for antimicrobial activity against 14 foodborne bacteria (Table 1). The antimicrobial sensitivity testing of the extracts was carried out on Mueller-Hinton Agar (MHA) plates by agar well diffusion assay as described before [38] with slight modification by following the Clinical and Laboratory Standards Institute guidelines [39]. In brief, wells measuring 6 mm in diameter were dug on the MHA plates using a sterile cork borer. The MHA plates were swabbed with the standard inoculum (turbidity adjusted to 0.5 McFarland standard, corresponding to approximately 1.5 ×10^8^ cfu/mL) of the test isolates. The RRP extract (100 µL) was transferred onto the corresponding wells. Autoclaved distilled water was used as a negative control. The plates were incubated at 37 °C for 24 h, and the zone of inhibition (ZOI) was reported in millimeters (mm).

#### 2.5.2. Determination of Minimum Inhibitory Concentration (MIC) and Minimum Bactericidal Concentration (MBC)

The ethanol RRP extract exhibited strong antimicrobial activity; therefore, the MIC and MBC were determined against two Gram-positive (*S. aureus* ATCC 25923, *L. monocytogenes* ATCC 19111) and two Gram-negative bacteria (*S.* Typhimurium ATCC14028 and *E. coli* ATCC 25922). In brief, 100 µL of sterile cation-adjusted Mueller Hinton broth (CAMHB) was added to each of the wells of a 96-well microtiter plate. To the first well of each row, 100 µL of ethanol extract (at a concentration of 200 mg/mL) was added and diluted serially in a two-fold dilution at concentrations ranging from 100–0.00038 mg/mL. The 0.5 McFarland standard suspension of a freshly grown log-phase bacterial culture of the test isolates was diluted 100X to a density of 10^6^ cfu/mL (9.9 mL CAMHB + 0.1 mL 0.5 McFarland standard suspension). Then, 10 µL of the diluted inoculum was transferred to each well within 30 min of preparation. Two wells of the plate were used as broth control (without culture) and culture control (without antibiotic), respectively. The plates were covered with aluminum foil and incubated at 37 °C for 18–24 h. To each well, resazurin (0.015% *w*/*v*) was added (30 µL/well), and further incubated for 2–4 h to observe color change. The lowest concentration of RRP extract that produced no color change (the blue color of resazurin remaining unchanged) was considered as the MIC value.

The MBC of the RRP extract was determined at the end of the incubation period by taking two 10 µL samples from each well showing no visible growth and plating the samples onto MHA plates. The resultant colonies were counted after overnight incubation at 37 °C for 18–24 h. The lowest dilution of the extract that produced at least 99.9% killing of the initial inoculum was considered the MBC value. The isolates treated with ciprofloxacin (512–0.25 mg/mL) were positive control.

The bactericidal or bacteriostatic activity was ascertained by calculating the MBC/MIC ratio or MIC index or tolerance level. Generally, a low MBC/MIC ratio (≤4) indicates that the antimicrobial action is bactericidal, i.e., it can kill the bacteria at concentrations closer to or equal to MIC. A high MBC/MIC ratio (≥4) indicates bacteriostatic, i.e., the antimicrobials can only inhibit the growth of bacteria but cannot kill them at concentrations closer to or equal to MIC.When the MBC/MIC ratio is very high (≥16), it indicates that the antimicrobial is tolerant, i.e., it has no bactericidal activity even at high concentrations [40,41].

#### 2.5.3. Dose and Time-Dependent In-Vitro Growth Kinetics of *S. aureus* and *L. monocytogenes* Treated with RRP Extract

The growth kinetics of the bacterial cultures were evaluated against RRP extract (at 1X MIC and 1X MBC) by incubating the log-phase grown bacterial cultures (inoculum adjusted to 1 × 10^7^ cfu/mL) in CAMHB. In brief, the log phase culture of each isolate was prepared by inoculating a single colony in BHI broth and incubating at 37 °C for 4 h in a shaker incubator (REMI, CIS-24 Plus, Mumbai, India). After incubation, the culture was pelleted by centrifugation (2370× *g* for 5 min) and resuspended in sterile PBS to attain a turbidity equivalent to 0.5 McFarland standard, which was further diluted to a concentration equivalent to 1 × 10^7^ CFU/mL using sterile CAMHB. From this dilution, 50 µL of inoculum was transferred to respective tubes containing 1X MIC and 1X MBC of RRP extract and incubated at 37 °C. The isolates treated with ciprofloxacin (1X MIC and 1X MBC) served as treatment control, while the target bacterial isolate in CAMHB served as untreated control. The aliquots of 10 µL collected at each fixed time interval (0, 3 h, 6 h, 9 h, 12 h, 24 h) were serially diluted in sterile PBS, plated on MHA plates, incubated at 37 °C for 24 h, enumerated and reported log10 cfu/mL. To ascertain the dose and time-dependent growth kinetics, a line chart was prepared using the mean (n = 6) bacterial colony counts (log_10_ cfu/mL) and time (hours) for each treatment.

#### 2.5.4. Stability Assessment

The RRP extracts were exposed for 30 min to varying temperatures (37 °C, 70 °C and 100 °C) and pH (2, 4, 6, and 8) conditions, and the refrigerated temperature (4 ± 1 °C) for 0, 15, and 30 days. Then, the antimicrobial activity was evaluated against *S. aureus* ATCC25923 and *L. monocytogenes* ATCC19111 using an agar well diffusion assay as above.

### 2.6. Effect of RRP on Lactobacilli Growth

The effect of RRP extract was tested on beneficial probiotic lactobacilli (*Lactiplantibacillus plantarum* MTCC 2621 and *Lacticaseibacillus casei* MTCC 1423), which are generally present as commensals in the gut by plate count method [42]. Briefly, the lactobacilli probiotic bacteria (100 µL each, 10^7^ CFU/mL) were inoculated in 100 µL of MRS broth medium containing RRP extract (1X MIC and 2X MIC for *S. aureus*) in a 96-well microtiter plate. The untreated bacterial cultures and MRS broth were used as positive and negative controls. After incubation at 37 °C for 48 h, the cultures were diluted with MRS broth, and 10 µL from each was plated on MRS agar plates, and colonies were counted.

### 2.7. Effect of RRP on Red Blood Cells

An in vitro hemolysis assay using sheep red blood cells (RBCs) was conducted to ascertain cytotoxicity, if any, against mammalian cells, as described before [43] with slight modification. Briefly, the aseptically collected sheep blood was washed several times with phosphate buffer saline (PBS) until a clear supernatant was obtained, centrifuged at 1000× *g* for 15 min, and resuspended in 10% (*v*/*v*) PBS containing 10 mM Dithiothreitol (DTT). Different concentrations (1X, 2X, 3X, 5X and 10X MIC for *S. aureus*) of 100 µL of RRP extract were added to individual tubes containing 100 µL of the sheep RBCs. Sheep RBCs (100 µL) with 100 µL each of 0.2% Triton X-100 and sterile PBS were used as positive control and negative control or blank, respectively. The reaction mixtures were incubated at 37 °C for 1 h and then centrifuged at 1300× *g* for 10 min. The supernatants were transferred to a 96-well microtiter plate, and the hemoglobin released was measured spectrophotometrically (Spectra Max^®^ M5, USA) at 540 nm. The hemolysis % was then calculated by using the following formula:Hemolysis (%) = (Absorbance of Sample − Absorbance of PBS)/(Absorbance of Triton X − Absorbance of PBS) × 100

### 2.8. Application of RRP in Meat-Model System

The application of RRP extract on food safety and quality was assessed using emulsion-based chicken meatballs containing RRP at 1% (T1) (10 mg/g, i.e., approximate 8X MIC and 7X MBC for *L. monocytogenes*, and 9X MIC and 8X MBC for *S. aureus*) and 2.5% (T2) (25 mg/g, i.e., 1X MIC or MBC for *S*. Typhimurium and *E. coli*; 9X MIC and 8X MBC for *L. monocytogenes*, 10X MIC and 9X MBC for *S. aureus*).The meatballs were prepared and cooked as described before [44] and stored at 4 ± 1 °C for 20 days under aerobic packaging conditions. Briefly, the frozen (−18 ± 2 °C) deboned chicken meat was thawed in a refrigerator (4 ± 1 °C), minced once through 8 mm and then through 4 mm plate of a meat grinder/chopper and then randomly separated into three batches to form three experimental groups. For each batch, meat, ice flakes, salt, phosphate, sodium nitrite, and vegetable oil were homogenized in a bowl chopper to form a fine emulsion. Spices, condiments, and RRP (at 1% level for T1 and at 2.5% level for T2) were then added to the emulsion in the final run of the minute and emulsified for 2–3 min until the final meat batter temperature reached 15 °C. The batch of emulsion without RRP was considered as control. Each batch of meat emulsion (500 g) was transferred to a tray to obtain individual meatball of around 20 g weight. All samples were steam cooked at 80 °C for 30 min until internal temperature reached 75 ± 1 °C. The cooked chicken meatballs were cooled to room temperature and were analyzed for various quality parameters at 5-day intervals. The meatball formulation is given in the Appendix A.

For sensory evaluation, about 30 g (at least 2 bites) of samples were pre-warmed for 20 s in a microwave oven and analyzed for different sensory attributes i.e., color and appearance, flavor, texture and tenderness, juiciness and overall acceptability, using an 8-point hedonic scale, where 8 = extremely desirable and 1= extremely undesirable by ten trained panelists in the late morning (10:30 a.m.–11:30 a.m.) [45]. Coded meatball samples along with plain potable water for mouth rinsing in between samples were provided to individual panel member in separate booths.

The oxidative changes were determined by analyzing 2-thiobarbituric acid reactive substances (TBARS) [46]. About 10 g of each meatball was homogenized with 25 mL of 20% precooled trichloroacetic acid (TCA) for 2 min using an Ultra Turrax tissue homogenizer (Model IKA^®^T 18, Janke and Kenkel, IKA Labor Technik, KG Janke & Kunkel-Str. 10, Staufen, Germany). The homogenate was filtered through a Whatman no. 1 filter paper (GE Healthcare, Waukesha, WI, USA) to obtain the TCA extract. About 3 mL of TCA extract was mixed well with 3 mL of 5 mM thio-barbituric acid (TBA) reagent in a separate test tube. The test tube containing the reaction mixture was placed in a water bath at 70 °C for 35 min and then cooled under tap water. The blank was prepared using equal volumes of TBA and TCA. The absorbance of the reaction mixture was recorded against a blank at 532 nm using a UV-visible spectrophotometer (Spectra Max^®^ M5, USA). The TBARS value of each sample was calculated by multiplying the absorbance value with a factor of 5.2 and expressed as mg malonaldehyde per kg (mg MDA/kg) of the sample.

The microbiological quality, including total viable counts (TVC), psychrophilic counts, *Escherichia coli* counts, and *Staphylococcus aureus* counts, were estimated as outlined by the American Public Health Association [47]. Briefly, a 10 g meat ball sample was transferred aseptically into a stomacher bag (Seward Medical, Newport Gwent, Wales, UK) containing 90 mL of 0.1% sterile buffered peptone water (BPW) (Hi-Media, Mumbai, India) and homogenized for 2 min in the stomacher (Lab Blender 400, Seward Medical, Newport Gwent, Wales, UK) to achieve 10^−1^ dilution. To prepare 10^−2^ dilution, 1 mL from 10^−1^ dilution was transferred into a test tube containing 9 mL of 0.1% sterile BPW and mixed to obtain 10^−3^ dilution and so forth. Serial dilutions were made with BPW under aseptic conditions. One milliliter aliquot from each serially diluted suspensions of the individual meatball sample homogenate was poured on the sterile petri plates and mixed in a swirling manner with the respective molten agar (45–50 °C) media (Hi-Media, Mumbai, India). Plate count agar (PCA) was used for TVC and Psychrophilic count. In contrast, Eosin Methylene Blue (EMB) agar and Baird Parker Agar (BPA) were used to enumerate *E. coli* and *S. aureus*, respectively. The plates were kept undisturbed until media got solidified and then incubated under aerobic conditions at 37 ± 1 °C for 48 h and 24 h for TVC and *E. coli* counts and *S. aureus* counts, respectively. However, plates were incubated at 4 ± 1 °C for 7 days in an inverted position for psychrophilic counts. Bacterial colonies were counted and the average number of colonies was multiplied by the reciprocal of the dilution and expressed as log_10_ cfu/g of the sample.

### 2.9. Statistical Analysis

The samples were analyzed thrice in duplicates (n = 6), except for sensory evaluation (n = 30), and the statistical analysis (one-way and two-way ANOVA) was performed using SPSS software (Version 20, IBM, Armonk, NY, USA). The data were presented as mean ± standard error (SE) using Duncan’s multiple range test, and the statistical significance was determined at a 95% confidence interval level (*p* < 0.05). Graphs were generated using GraphPad Prism 9.3 (Boston, MA, USA).

## 3. Results and Discussion

### 3.1. Characteristics of Ethanol and Water Extracts of Ratanjot (A. tinctoria L.) Root

The total extractable components (TECs) of soluble fractions by ethanol was 18.27%, while the microwave-assisted hot water (MAHW) method yielded 6.29% (Table 2). This suggests that RRP was more soluble in alcohol and partially soluble in water. Using a water-infusion method, Guemmaz et al. [48] reported a similar phenolic compound yield of 7.25% from *A. tinctoria* L. aerial parts. Alcohol-based phenolic compound extraction has been well documented in the literature. Using methanol, Gao et al. [49] obtained about 16.5% and 15.9% from the inner bark and outer bark of Port Oxford Cedar, respectively. Likewise, Ngo et al. [50] reported a 15.6% yield using absolute methanol from *Salacia chinensis* L. root.

The mean pH values of ethanol and MAHW extracts were 2.92 and 3.15, respectively (Table 2). Zannou and Koca [25] reported similar pH values of *A. tinctoria* L. root extracts using natural deep eutectic solvents like sodium acetate/lactic acid (pH = 3.41) and sodium acetate/formic acid (pH = 3).The acidic nature of the *A. tinctoria* L. root extract might also contribute to its biological activity and be suitable for use in low-acid foods.

The mean values for the standard CIELAB color parameters for ethanol and MAHWextracts are summarized in Table 2, and the respective values are *L** (25.07 and 56.55), *a** (45.13 and 12.02), *b** (63.35 and 62.20), H°_ab_ (2.61 and 41.70) and C*_ab_ (77.97 and 63.72). In general, the values of *L** range from 0–100 for black and white respectively: *a** values correspond to negative values for greenness and positive values for redness, *b** values correspond to negative values for bluish color and positive values for yellowness, h_ab_ correlate to tonality and C*_ab_ correlate to color vividity. The ethanol extract had significantly (*p* < 0.05) lower *L** and h_ab_ and higher *a** values than the MAHW extract, indicating that the ethanol extract has a dark red color compared to the light red and brighter color tone for water extract, thus making RRP extract a highly valued colorant for use in food, and the cosmetic, textile and pharmaceutical industries.

### 3.2. Antioxidant Activity of RRP

#### 3.2.1. Total Phenolic Content

Plant phenolic compounds are free radical scavengers, and a plant extract’s total phenolic content (TPC) can be correlated to its antioxidant activity [51]. The mean TPC (mg GAE/g) of ethanol extract of RRP was 45.10 and 4.57 for MAHW extract (Figure 2). The difference in TPC of both the extracts might be because most of the phytochemicals, including phenolic compounds of RRP, were more soluble in alcohol and partially soluble in hot water.

These findings were partially corroborated by that of Zannou and Koca [25], who found that the total phenolic contents of 70% ethanol and 80% methanol extract of *A. tinctoria* root were 49.70 mg GAE/g and 33.10 mg GAE/g, respectively. However, in contrast to our results, these authors reported a higher total phenolic content, i.e., 86.80 mg GAE/g of water extract of *A. tinctoria* L. root. Nevertheless, much higher values for mean total phenolic contents were obtained for *A. tinctoria* L. root extracts using natural deep eutectic solvents, like sodium acetate/lactic acid (170.96 mg GAE/g) and sodium acetate/formic acid (175.97 mg GAE/g). Our findings partially agree with the reports of Kozłowska et al. [52], who found that the total phenolic content of the *Angelica archangelica* L. root extract was 20.35 mg GAE/g. Similarly, Gao et al. [49] also found that the mean total phenolic contents of bark and wood powders of Port Oxford Cedar ranged from 2.56–88.76 mg GAE/g. Other research similarly documented that the total phenolic contents of water extracts prepared from *A. tinctoria* L. aerial parts by decoction, maceration, and infusion were 173.28, 144.50 and 189.66 mg GAE/g, respectively. Another study also reported that the total phenolic content of the red radish (*Raphanus sativus* L.) root was 341.45 mg GAE/100g [53].

The differences in total phenolic contents observed by different researchers can be attributed to a variety of factors, such as plant variety, plant parts, climate, geographical location, growth conditions, extraction method, and kind of solvents used [54].

#### 3.2.2. 2,2-Diphenyl-1-Picrylhydrazyl (DPPH) Radical Scavenging Assay

The DPPH assay measures the electron-donating activity of a substance to scavenge or reduce DPPH free radicals and measures its antioxidant activity [55]. The data presented in Figure 2 shows a significant difference (*p* < 0.05) in mean IC_50_ values of ethanol extract (76.86 µg/mL) and MAHW extract (170.60 µg/mL) of RRP in scavenging DPPH free radicals as compared to the standard antioxidants like BHT (13.12 µg/mL) and BHA (6.99 µg/mL). However, BHA and BHT showed much lower IC_50_ concentrations than both extracts, which indicates their potency as standard antioxidants. Moreover, among both types of RRP extracts, 50% scavenging of the DPPH free radicals was more efficiently performed by the ethanol extract than the MAHW extract. This might be because more phenolic compounds with DPPH radical scavenging activity were soluble in ethanol than in water.

Our results are also in agreement with the findings of Zannou and Koca [25], who estimated the mean IC_50_ values for DPPH free radical scavenging activity of *A. tinctoria* L. root using 70% ethanol extract (75.24 mmol Trolox Equivalent or TE/g), 80% methanolic extract (74.07 mmol TE/g), water extract (97.39 mmol TE/g), extracts using natural deep eutectic solvents like sodium acetate/lactic acid (112.31 mmol TE/g) and sodium acetate/formic acid (130.91 mmol TE/g). Likewise, Guemmaz et al. [48] also reported that the mean IC_50_ values in the DPPH radical scavenging assay for water extracts prepared from *A. tinctoria* L.aerial parts by decoction, maceration, and infusion were 0.17 mg/mL, 1.006 mg/mL and 0.09 mg/mL, respectively. In another experiment, Gao et al. [49] also documented that the mean IC_50_ values of heartwood, sapwood, inner bark, and outer bark extracts of Port Oxford Cedar were 64.77 mg/mL, 29.03 mg/mL, 10.31 mg/mL, 19.87 mg/mL, respectively. Contrastingly, Ngo et al. [50] reported a very high IC_50_ value (407.47 mmol TE/g) for acetone (50%) extracts of *S. chinensis* L. root in DPPH radical scavenging. The differences in results of the DPPH assay might be due to various factors, as mentioned earlier, regarding total phenolic contents. From the above results, it can be deduced that there is a direct correlation between the total phenolic content of a plant extract and its DPPH free radical scavenging activity. Typically, plant extracts that contain more phenolic hydroxyl groups demonstrate greater DPPH radical scavenging activity [56].

#### 3.2.3. 2.2-Azino-Bis (3-Ethylbenzothiazoline-6-Sulfonic Acid) (ABTS^.+^) Radical Cation Decolorization Assay

The ABTS assay is a simple, convenient, and efficient measure of total antioxidant capacity, which determines the ability of a test substance to neutralize the bluish-green ABTS free radical. The significant (*p* < 0.05) differences in IC_50_ concentrations for ethanol extract (37.95 µg/mL), MAHW extract of RRP (57.09 µg/mL), BHT (6.96 µg/mL), and BHA (5.06 µg/mL) in ABTS assay were evident (Figure 2). Significantly (*p* < 0.05) higher ABTS radical scavenging activity and lower IC_50_ concentrations were obtained for ethanol extract than the water extract of RRP, possibly due to better solubility and extractability of phenolic components in alcohol. Although more potent ABTS radical scavenging activities were observed for the standard chemical antioxidants, like BHT and BHA, the observed IC_50_ concentrations of RRP extract in ABTS assay are highly desirable because of their natural origin and health benefits.

Like our findings, Sundaram et al. [57] reported that the IC_50_ for ABTS radical scavenging of *Strobilanthes heyneanus* root extracts from Southwest India was 33.92 µg/mL. These findings bear similarity with the reports of Kozłowska et al. [52], who reported that the IC_50_ value in ABTS assay for *Angelica archangelica* L. root extract was 86.54 µg/mL. In contrast to our results, a very high IC_50_ value (414.38 mmol TE/g) or lower ABTS radical scavenging activity for acetone (50%) extract of *S. chinensis* L. root was observed by Ngo et al. [50]. A recent study by Ganos et al. [58] found that the IC_50_ concentration of methanol extract of *A. tinctoria* L.aerial part was 366.88 mg TE/g. Similarly, the factors mentioned earlier might also contribute to the differences in results obtained for ABTS radical scavenging activity. The findings from the DPPH and ABTS radical scavenging assays in this study indicate the potent antioxidant properties of the RRP. Generally, phenolic hydroxyl groups found in plants, such as catechol and hydroquinone, donate electrons to neutralize DPPH free radicals, while phenols, flavonoids, and carotenoids are crucial for ABTS radical scavenging [59]. Both assays aid in identifying potent natural antioxidants valuable for the food and pharmaceutical industries.

#### 3.2.4. Identification of Phytochemicals Using Gas Chromatography–Mass Spectrometry (GC–MS/MS)

Understanding the phytochemical compounds in Ratanjot (*A. tinctoria* L.) root extract is essential for assessing its bioactivity and optimizing its application in the food and pharmaceutical industry [54]. The GC–MS/MS chromatography revealed the presence of three major phenolic compounds: (i) butanoic acid, 3-hydroxy-3-methyl-; (ii) arnebin 7; and (iii) diisooctyl pthalate (Figure 3). Butanoic acid, 3-hydroxy-3-methyl- is also known as butyric acid, 3-hydroxy-3-methyl- and is soluble in alcohol. Butyric acid derivatives have been identified in marine red algae *Rhodomela confervoides* [60,61] and Mexican corn products [62] that contribute to potent DPPH and ABTS radical scavenging activities. Butyric acid contributes to overall health through anti-inflammatory and digestive functions [60]. Butyric acid also contributes flavor and aroma in dark chocolate [61] and coffee silver skin [60].

Arnebin 7, also known as deoxyshikonin, is insoluble in water but soluble in ethanol [63]. It is an enantiomer of alkannin and is grouped under hydroxynaphthoquinones [60], which are DNA topoisomerase inhibitors [27]. Alkannin and Shikonin are known for their antimicrobial [64] and healing properties. They are active ingredients in therapeutic wound healing ointments, like hexiderm, histoplastin red, etc. [27].

Diisooctyl pthalate (DIOP), technically known as pthalic acid, bis-isooctyl ester is insoluble in water but soluble in organic solvents. DIOP from marine brown algae *Sargassum wightii* was found to be an antimicrobial compound [65]. Similarly, DIOP was also identified as a phytochemical in *Vernonia glabra* root ethyl acetate extracts, which showed antitubercular activity against *Mycobacterium tuberculosis* [66].

In addition, researchers have also reported the presence of other bioactive components such as alkannins, arnebifuranone, alkandiol, napthoquinones, alkanfuranol and shikonin in *A. tinctoria* L. root extract [25,26,67].

### 3.3. Antimicrobial Activity of RRP

#### 3.3.1. Antimicrobial Activity Spectrum of RRP

Antimicrobial activities of ethanol and MAHW extracts against several Gram-positive and Gram-negative bacterial pathogens were tested by agar well diffusion assay (Table 3). The ethanol extract was highly inhibitory against both Gram-positive *S. aureus* ATCC 25923, *L. monocytogenes* ATCC 19111, *L. monocytogenes* ATCC 13932, and some field isolates of *S. aureus* and Gram-negative bacteria *S.* Typhimurium ATCC 14028 and *E. coli* ATCC 25922. In contrast, the water extract had no antimicrobial activities. This might be because the phytochemicals responsible for the antimicrobial activity of RRP were soluble in ethanol, not in water. Other researchers also obtained similar results and observed that ethanol extract of *A. tinctoria* L. roots had a more potent antimicrobial effect against Gram-positive bacteria than Gram-negative bacteria and the water extracts had no or negligible effect. Interestingly, the Gram-positive bacteria were more sensitive than the Gram-negative bacteria, which agrees with other reports [27,31,64].

#### 3.3.2. Determination of MIC and MBC

Next, we determined the MIC and MBC values of the ethanol extract. It is pertinent to mention here that no MAHW extract of RRP was used in subsequent experiments, since it had no detectable antimicrobial activity. The MIC of ethanol extract for *S. aureus* ATCC 25923 and *L. monocytogenes* ATCC 19111 was 0.049 and 0.098 mg/mL, respectively, while this was 25 mg/mL for *Salmonella* and *E. coli*. The MBC for *S. aureus* ATCC 25923 was 0.098 mg/mL, while this was 0.195 mg/mLfor *L. monocytogenes* ATCC 19111. Furthermore, the MIC and MBC concentrations of ciprofloxacin against Gram-negative bacteria were 0.125 µg/mLand 0.250 µg/mL, respectively, whereas against Gram-positive bacteria these were 0.250 µg/mLand 0.500 µg/mL, respectively.

The tolerance levels for RRP extract were 1 mg/mL and 2 mg/mL for Gram-negative and Gram-positive bacteria, respectively. Similarly, a tolerance level of 2 mg ciprofloxacin was recorded for all four target bacteria (Table 4). The ratio of MBC/MIC ≤ 4 suggested that the RRP extract had bactericidal activity, similar to ciprofloxacin, i.e., RRP preparation can kill the target pathogens at concentrations closer to or equal to MIC [40,41,44]. However, slightly higher concentrations of MIC (6.25–25 mg/mL) and MBC (>100 mg/mL) were reported by Khan et al. [31] for ethanol, chloroform, hexane, and aqueous extracts of *A. tinctoria* L. leaves against multidrug-resistant (MDR) human pathogens. The difference between our results and the study of Khan et al. [31] suggests that the roots of *A. tinctoria* L. possess higher antimicrobial activities than the leaves. Another study reported that the MIC of the ethanol extract of *A. tinctoria* L. roots against *B. subtilis* and *S. aureus* was 2 µg/mL, whereas the MIC for *E. coli* and *Pseudomonas aeruginosa* was 4 µg/mL [64].

#### 3.3.3. Effect of RRP Extract on Growth Kinetics of *S. aureus* and *L. monocytogenes*

We also determined the growth kinetics of *S. aureus* and *L. monocytogenes* for 24 h after exposure to RRP extract (1X MIC and 1X MBC) (Figure 4). RRP extract (1X MIC) caused about a 5-log reduction of *S. aureus* within 6 h with a final reduction of 4 logs in 24 h, while 1X MBC caused about 5 log reduction in 6 h and was undetectable after 9 h. Ciprofloxacin (both 1 X MIC and MBC) used as a positive control showed similar growth kinetics as RRP extract against *S. aureus* (Figure 4A). A similar growth kinetics of *L. monocytogenes* was observed in the presence of RRP extract at dose levels of 1 X MIC and MBC (Figure 4B). Interestingly, ethanol extract of *Erythrina caffra* Thumb. tree bark at 2X MIC killed several pathogens, including *Micrococcus luteus*, *Proteus vulgaris*, and *S. aureus*, after 8 h of exposure [40]. Our data indicate that RRP extract is highly bactericidal, causing a 5-log reduction of both tested pathogens within 6 h.

#### 3.3.4. Stability of RRP Extract under a Wide Range of Conditions

Considering the potential application of RRP in food, we tested its stability in different food-related environments, including a broad range of pH (2–8) and temperatures (4–100 °C), and the antimicrobial activity against *Listeria monocytogenes* ATCC 19111 and *Staphylococcus aureus* ATCC 25923 (Figure 5).

RRP was highly inhibitory to both pathogens when maintained at pH 2–6; however, it lost its activity at pH 8. Potential factors contributing to the diminished antibacterial activity of acidic RRP at pH 8 could be the degradation of bioactive compounds, changes in ionization states leading to reduced solubility of active compounds, and decreased interaction with bacteria under alkaline conditions [68,69]. RRP remained active at a broad temperature range, with maximum activity observed at 37 °C, followed by a gradual and significant (*p* < 0.05) decrease at 70 °C and 100 °C. Furthermore, RRP was held at 4 °C for 30 days and maintained its inhibitory activity against both pathogens. These data demonstrate that RRP is highly stable at a broad range of pH and temperature associated with food product formulations and storage, thus suitable for food preservation [60]. Our findings bear similarity to the findings of both these studies and suggest that RRP has excellent thermostability at moderate to high temperatures, pH stability at acidic conditions (pH = 2–6) and refrigerated stability for up to 30 days.

### 3.4. Effect of RRP on Lactobacillus Growth

The effect of Ratanjot (*Alkanna tinctoria* L.) root extract (RRP) on probiotic lactobacilli growth was monitored, since they are considered natural gut microflora. In the presence of RRP at 1X MIC and 2X MIC, the average counts of model probiotics, *Lactiplantibacillus plantarum* MTCC 2621 and *Lacticaeibacillus casei* MTCC 1423, were 9.3 log cfu/mL after 24 h exposure (Figure 6). Probiotic lactobacilli confer pathogen colonization resistance and exhibit potent anti-inflammatory and immunomodulatory responses and gut health [70,71,72]. Our results indicate that RRP extract has no apparent inhibitory effect on tested probiotic lactobacilli; thus, the probiotics can potentially maintain homeostasis and gut health.

Selective inhibition against the potentially harmful foodborne pathogenic bacteria and maintenance of probiotic gut lactobacilli is a crucial property of plant phenolics and is linked with the host’s gut health [73]. The antioxidant dietary fibers, flavonoids, and phenolic compounds present in some plants act as prebiotics for stimulating the growth of probiotic gut lactobacilli [74,75], making them ideal ingredients of human functional foods [76], ruminant diets and poultry feed [77]. Likewise, the non-inhibitory and growth-stimulating effects of several plant extracts, such as *Aloysia citrodora* leaves extract [78], *Achillea millefolium* L. or yarrow herb extract [74], *Cannabis sativa* L. or hemp seed extracts [73], and *Sesbania grandiflora* or agasti flower extract [79], on probiotic lactobacilli like *L. plantarum*, *Lactobacillus acidophilus*, *Lacticaseibacillus rhamnosus*, *Bifidobacterium bifidum* and probiotic yeast have been reported.

### 3.5. Effect of RRP on Red Blood Cells

In vitro, hemolysis assay is a simple and cost-effective alternative test to assess the safety of phytochemicals on mammalian cells [80], and this test is widely accepted for toxicity testing of molecules destined for food and/or medicine use [81]. The results show that RRP extracts at 1–3X MIC did not cause any hemolysis, while at 5X and 10 X MIC they caused a meager 0.1 and 0.3% hemolysis when compared with Triton-X 100, a non-ionic detergent known to cause hemolysis by disrupting the RBC membrane [82] (Figure 7). Determining hemolysis when RBCs come in contact with any absorbed or injected substance represents a potential for assessing the safety or toxicity of the substance [83]. If the degree of hemolysis caused by a plant extract is >30%, it is considered hazardous for RBCs [84]. The anti-hemolytic properties of plant extracts are attributed to their phenolic contents exerting antioxidant activity on RBC membrane rupture [85,86]. Similar anti-hemolytic activities of phytochemicals or extracts from several plants, like *Portulacaria afra* [84], *Albizia richardiana* [85],mangrove plants [86], and *Acacia hydaspica* [83], have been reported.

### 3.6. Application of RRP as Food Preservative in Meat-Model System

Chicken meatballs were prepared to contain RRP at 1% (T1) (10 mg/g, i.e., approximate 8X MIC and 7X MBC for *L. monocytogenes*, and 9X MIC and 8X MBC for *S. aureus*) and 2.5% (T2) (25 mg/g, i.e., 1X MIC or MBC for *S*. Typhimurium and *E. coli*; 9X MIC and 8X MBC for *L. monocytogenes*, 10X MIC and 9X MBC for *S. aureus*), cooked and stored at 4 °C for 20 days (Figure 8A). Antioxidant activity (TBARS values) as measured by the production of malonaldehyde (mg/kg) indicated a slower rate of increase in the TBARS value for T1 and T2 samples, which were well within the acceptable limit (1 mg malonaldehyde/kg). In contrast, the control meatballs exceeded the permissible limit (1.43 mg malonaldehyde/kg) (Figure 8B).

Analysis of the microbial counts of the control chicken meatballs exceeded the permissible limits for TVC (3–4 log_10_ cfu/g) (Figure 8C), psychrophilic (4.1 log cfu/g) (Figure 8D), *E. coli* (1–2 log_10_ cfu/g) (Figure 8E) and *S. aureus* (1–2 log_10_ cfu/g) counts (Figure 8F) as established by Food Safety and Standards Authority of India, 2023 [87].In contrast, microbial counts in both T1 and T2 meatball samples were within the acceptable limit and are perceived as microbiologically safe. Notably, the counts of psychrophilic bacteria, *E. coli* and *S. aureus* were undetectable in all samples on day 5; however, on day 10, control samples showed counts for all the three groups, while they were undetectable in T1 and T2 samples (Figure 8D–F).

The sensory scores obtained for the chicken meatballs formulated with RRP (Table 5) revealed that both the T1 and T2 samples received better scores for color and appearance, flavor, texture and tenderness, juiciness, and overall acceptability than the control meatballs. From Day 0, no significant differences (*p* < 0.05) were evident between the control and treated samples (T1 and T2), whereas, from Day 5 of storage, significant differences (*p* < 0.05) in color, appearance, flavor, juiciness, and overall acceptability scores were observed between T1 and T2, likely due to the higher concentration of RRP in T2 compared to T1. Throughout the storage period, the mean scores for color and appearance, flavor, texture and tenderness, and juiciness across all samples reflected the freshness, and extent of spoilage due to microbial activity, as well as the degradation of lipids, proteins, and moisture [88]. The mean overall acceptability scores for the control, T1, and T2 samples ranged from 6.84–4.02, 6.89–5.06, 6.92–5.13 on day 0 and day 20 of refrigerated storage. The mean scores for overall acceptability assessed sensory quality and shelf life, encapsulating the combined sensory experiences of all attributes [54]. The improvement in the sensory quality of treated meatballs could be attributed to the antioxidant, antimicrobial properties, and unique colorimetric characteristics of *A. tinctoria* L. roots. Similar improvements in sensory attributes of different meat products using plant parts are well documented by other researchers [44,89,90,91].

These results indicate that phenolic compounds and RRP’s potent antioxidant and antimicrobial activity were responsible for producing microbiologically safe ready-to-eat products. Our results agree with several other studies on using plant extracts and essential oils as food preservatives [44,79,92].

## 4. Conclusions

The current study contributes remarkably to the pre-existing, limited scientific reports on the antimicrobial and antioxidant activities of Ratanjot or *A. tinctoria* L. root extract. The ethanol extraction method was much superior to the hot water extraction, and the active components were identified as butanoic acid 3-hydroxy-3-methyl-, arnebin 7, and diisooctyl pthalate. The extract displayed potent antioxidant activity. It also had strong bactericidal activity against Gram-positive and Gram-negative bacterial pathogens, including *Staphylococcus aureus*, *Listeria monocytogenes*, *Salmonella enterica* serovar Typhimurium, and *Escherichia coli*. RRP extract achieved a 5-log reduction in *Staphylococcus aureus* and *Listeria monocytogenes* growth within 6 h. Surprisingly, antimicrobial activity is much more robust against Gram-positive than Gram-negative bacteria. Interestingly, the extract did not inhibit the growth of common probiotic lactobacilli, implying its selective protection of probiotics known to exert health benefits to the host. Besides, the extract did not cause appreciable hemolysis even at 10X MIC of *Staphylococcus aureus*, indicating it to be non-toxic, and its routine use in Asian cuisine suggests that the RRP is safe as a food preservative.

The incorporation of RRP at 1% (T1) (10 mg/g) and 2.5% (T2) (25 mg/g) levels in the meat-model system showed its effectiveness in retarding lipid oxidation and microbial counts under refrigerated storage for 20 days. The observed antioxidant and antimicrobial activities of Ratanjot (*A. tinctoria* L.) root extract could be attributed to the plethora of bioactive phytochemicals. The findings of the current study provide a scientific basis for the potential application of Ratanjot root extract in the food industry as a natural preservative for improving microbial quality and food safety, enhancing sensory qualities, retarding lipid oxidation, and extending shelf life.

## Figures and Tables

**Figure 1 foods-13-02254-f001:**
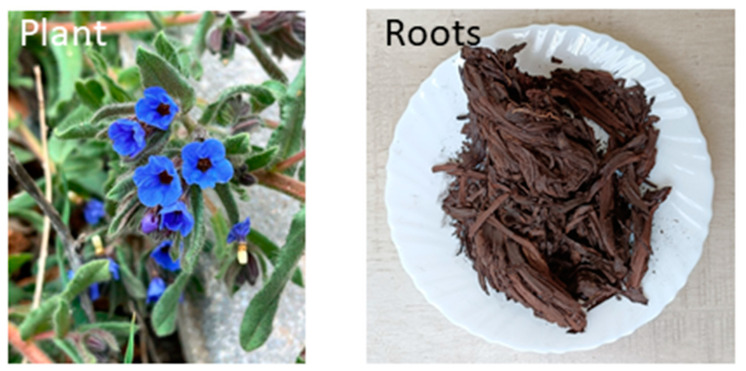
*Alkanna tinctoria* (known as Ratanjot) plant and roots.

**Figure 2 foods-13-02254-f002:**
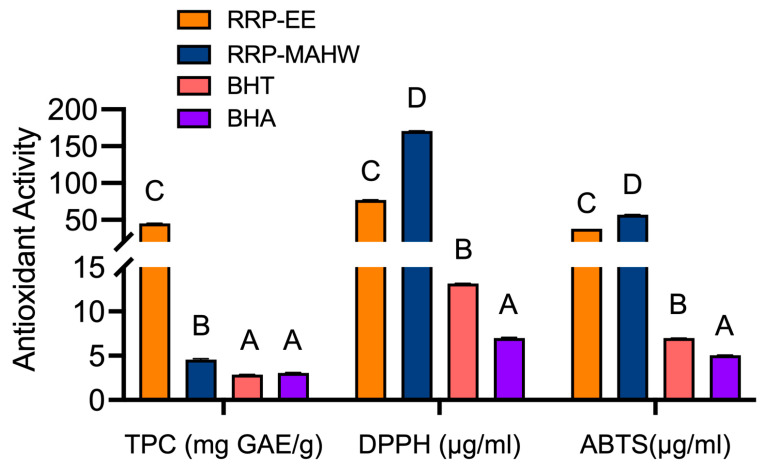
Antioxidant activity of ethanol (EE) and water extract (MAHW) of Ratanjot (*Alkanna tinctoria* L.) root (RRP) extract. Bars are average (n = 6) ± SE bearing different superscript(s) (A, B, C, D) differ significantly at *p* < 0.05. TPC, total phenolic compound; ABTS, 2.2-Azino-bis (3-ethylbenzothiazoline-6-sulfonic acid); DPPH, 2,2-Diphenyl-1-picrylhydrazyl (DPPH); GAE, gallic acid equivalent.

**Figure 3 foods-13-02254-f003:**
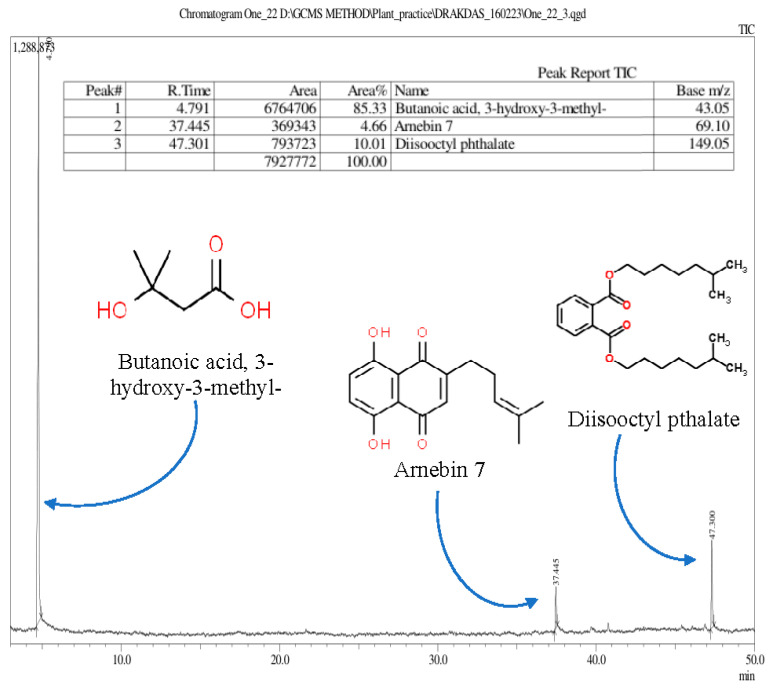
Gas chromatography and mass spectrometry analysis of *Alkanna tinctoria* L. (known as Ratanjot) root ethanol extract produced three major components: Butanoic acid, 3-hydroxy-3-methyl-; Arnebin 7, and Diisooctyl pthalate.

**Figure 4 foods-13-02254-f004:**
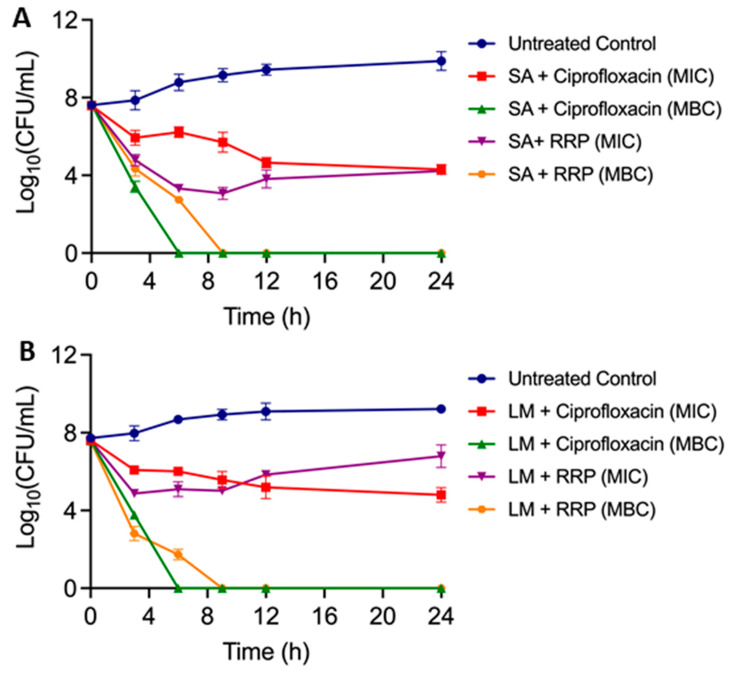
Growth kinetics of (**A**) *Staphylococcus aureus* (SA) ATCC 25923 and (**B**) *Listeria monocytogenes* (LM) ATCC 19111 after exposure to Ratanjot (*Alkanna tinctoria* L.) root extract (ethanol extract), RRP at 1X MIC and MBC for 24 h. Ciprofloxacin (1 X MIC and 1 X MBC) was used as a positive control.

**Figure 5 foods-13-02254-f005:**
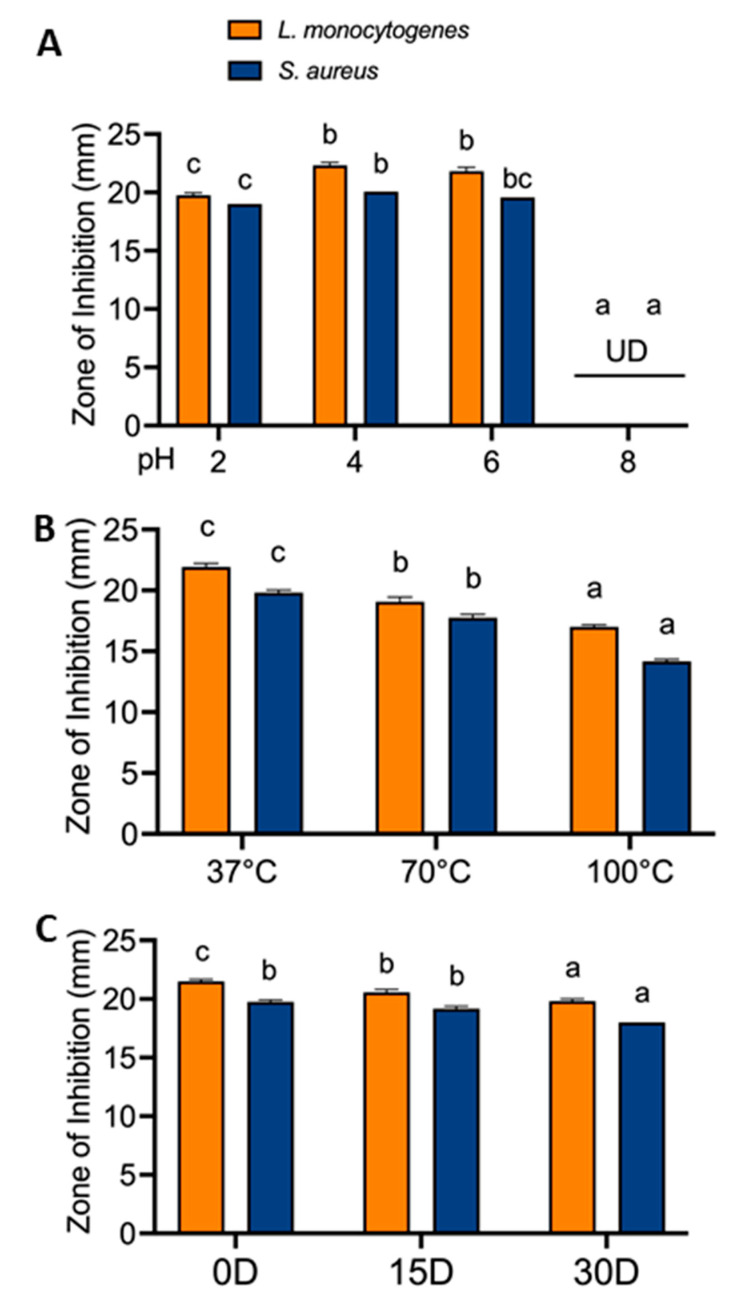
Antimicrobial activity of Ratanjot (*Alkanna tinctoria* L.) root extract (RRP) exposed to various (**A**) pH ranges 2–8, (**B**) temperature ranges (37–100 °C) and (**C**) Storage days (0–30 days) at 4 °C for 30 days against *Listeria monocytogenes* and *Staphylococcus aureus* analyzed by agar well-diffusion assay. Bars marked with different letters (a, b, c) are significantly different at *p* < 0.05. UD: Undetected.

**Figure 6 foods-13-02254-f006:**
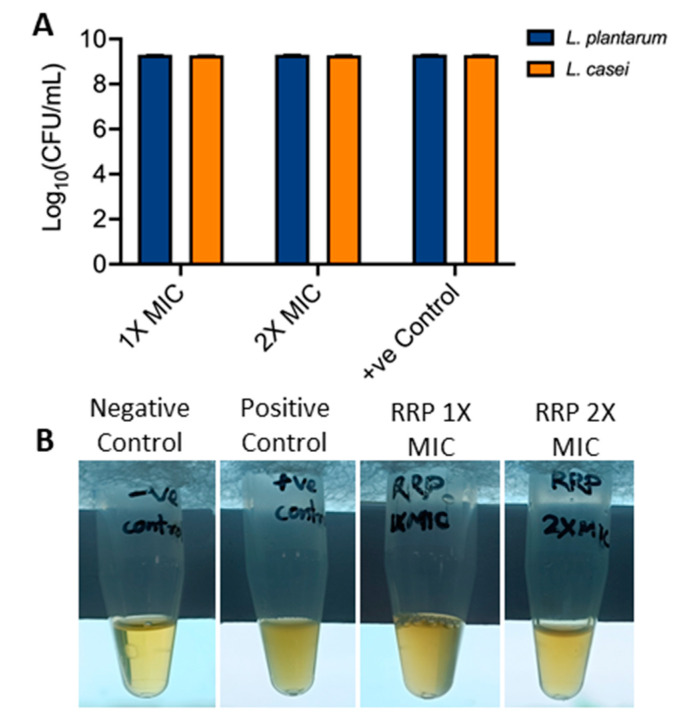
Effect of Ratanjot (*Alkanna tinctoria* L.) root extract (RRP) at 1X and 2X MIC on *Lactobacillus* growth. (**A**) *Lactiplantibacillus plantarum* and *Lacticaseibacillus casei* counts (log 10 CFU/mL) after 24 h in MRS broth, (**B**) visual examination of turbidity of *L. casei* growth after 24 h.

**Figure 7 foods-13-02254-f007:**
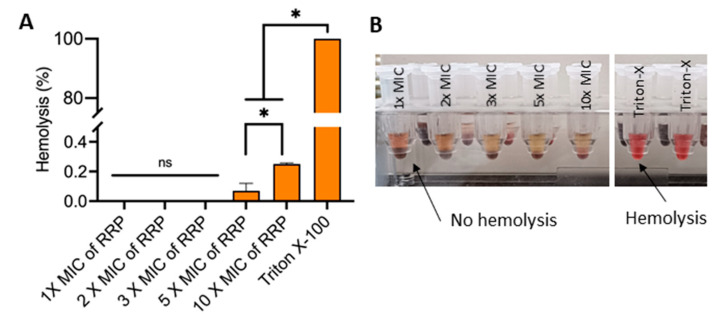
In vitro hemolysis assay of Ratanjot (*Alkanna tinctoria* L.) root extract (RRP) at various concentrations (1–10 XMIC). (**A**) Bar diagram showing percent hemolysis of sheep red blood cells. Data are mean (n = 6) ± SE. *, *p* < 0.05. Triton X-100 (0.1%) used as a positive control for complete lysis. (**B**) Photographs showing the difference in hemolysis and no hemolysis of RBC.

**Figure 8 foods-13-02254-f008:**
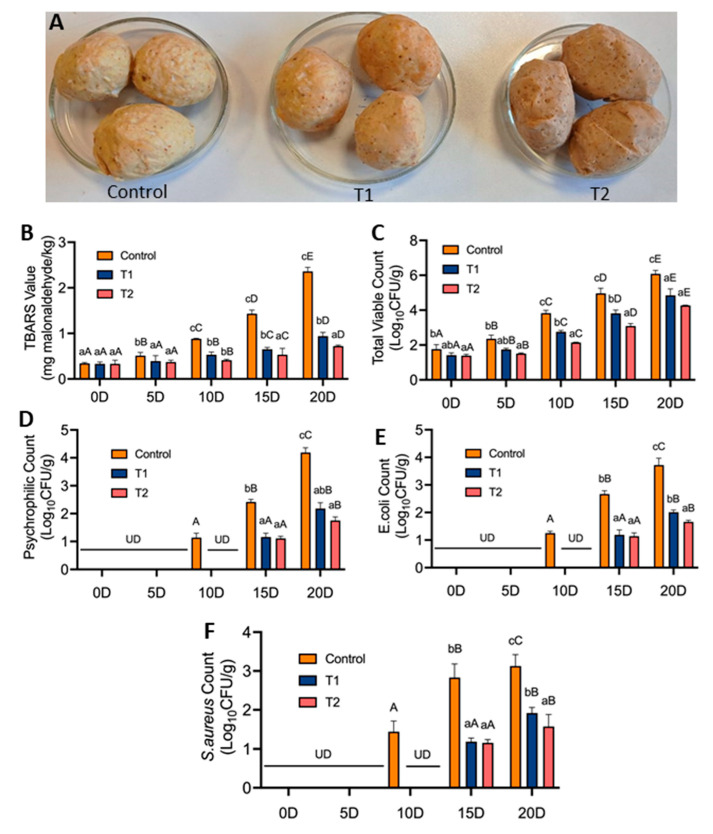
Application of Ratanjot (*Alkanna tinctoria* L.) root extract (RRP) at 1% (T1) and 2.5% (T2) in chicken meatballs on antioxidant (TBARS) and antimicrobial activity stored at 4 °C for 20 days. (**A**) Chicken meat balls incorporated with RRP. (**B**) TBARS values, (**C**) TVC, (**D**) Psychrophilic counts, (**E**) *E. coli* counts, and (**F**) *S. aureus* counts of control and RRP incorporated meat balls during refrigerated storage for 20 days. Bars marked with uppercase letters (A, B, C, D, E) signify differences in values on various days, and lowercase letters (a, b, c) signify differences within the same day for all three samples. *p* < 0.05. UD: Undetected.

**Table 1 foods-13-02254-t001:** Bacterial cultures used in this study.

Culture	Source
*Staphylococcus aureus* ATCC 25923	American Type Culture Collection (ATCC), Manasas, VA, USA
*Listeria monocytogenes* ATCC 19111	ATCC, Manasas, VA, USA
*Listeria monocytogenes* ATCC 13932	ATCC, Manasas, VA, USA
*Salmonella enterica* serovar Typhimurium ATCC 14028	ATCC, Manasas, VA, USA
*Escherichia coli* ATCC 25922	ATCC, Manasas, VA, USA
*Lactiplantibacillus plantarum* MTCC 2621	Microbial Type Culture Collection and Gene Bank (MTCC), Chandigarh, India
*Lacticaseibacillus casei* MTCC 1423	MTCC, Chandigarh, India
*Staphylococcus aureus* strain MZP/SM/05	Swine meat from Mirzapur, Uttar Pradesh, India
*Staphylococcus aureus* MZP/PM/38	Poultry meat from Mirzapur, Uttar Pradesh, India
*Staphylococcus aureus* MZP/GM/02	Goat meat from Mirzapur, Uttar Pradesh, India
*Staphylococcus aureus* ALD/PM/22	Poultry meat from Allahabad, Uttar Pradesh, India
*Staphylococcus aureus* ALD/PM/23	Poultry meat from Allahabad, Uttar Pradesh, India
*Staphylococcus aureus* LKO/GM/14	Goat meat from Lucknow, Uttar Pradesh, India
*Staphylococcus aureus* LKO/PM/04	Poultry meat from Lucknow, Uttar Pradesh, India
*Staphylococcus aureus* LKO/PM/08	Poultry meat from Lucknow, Uttar Pradesh, India
*Staphylococcus aureus* LKO/GM/11	Goat meat from Lucknow, Uttar Pradesh, India

**Table 2 foods-13-02254-t002:** Characteristics of ethanol and microwave-assisted hot water (MAHW) extracts.

Parameter	Ethanol Extract ± SE *	MAHW Extract ± SE *
Total Phenolic Content (TPC)%	18.27 ± 0.69 ^b^	6.29 ± 0.30 ^a^
pH	2.92 ± 0.02 ^a^	3.15 ± 0.01 ^b^
CIELAB color parameters		
*L**	25.07 ± 0.97 ^a^	56.55 ± 0.79 ^b^
*a**	45.13 ± 0.44 ^b^	12.02 ± 0.76 ^a^
*b**	63.35 ± 0.53	62.20 ± 0.65
H°_ab_	2.61 ± 0.08 ^a^	41.70 ± 0.31 ^b^
C*_ab_	77.97 ± 0.61	63.72 ± 0.38

* SE, Standard Error. Values labeled with superscripts (a, b) in the same row are significantly different at *p* < 0.05.

**Table 3 foods-13-02254-t003:** Antimicrobial activities of RRP extracts against some foodborne pathogens by agar well diffusion assay.

Bacteria	Zone of Inhibition (mm) (Mean ± SE *)	
	Ethanol Extract	Hot Water Extract
*Staphylococcus aureus* ATCC 25923	19.75 ± 0.11 ^ef^	0.00 ± 0.00
*Listeria monocytogenes* ATCC 13932	21.17 ± 0.28 ^gh^	0.00 ± 0.00
*L. monocytogenes* ATCC 19111	21.92 ± 0.33 ^h^	0.00 ± 0.00
*Salmonella* Typhimurium ATCC 14028	9.92 ± 0.15 ^b^	0.00 ± 0.00
*Escherichia coli* ATCC 25922	8.58 ± 0.24 ^a^	0.00 ± 0.00
*S. aureus* MZP/SM/05	21.17 ± 0.40 ^gh^	0.00 ± 0.00
*S. aureus* MZP/PM/38	18.25 ± 0.11 ^d^	0.00 ± 0.00
*S. aureus* MZP/GM/02	20.08 ±0.27 ^fg^	0.00 ± 0.00
*S. aureus* ALD/PM/22	19.50 ± 0.26 ^ef^	0.00 ± 0.00
*S. aureus* ALD/PM/23	18.67 ± 0.25 ^de^	0.00 ± 0.00
*S. aureus* LKO/GM/14	27.50 ± 0.22 ^i^	0.00 ± 0.00
*S. aureus* LKO/PM/04	16.00 ± 0.13 ^c^	0.00 ± 0.00
*S. aureus* LKO/PM/08	20.00 ±0.37 ^fg^	0.00 ± 0.00
*S. aureus* LKO/GM/11	27.83 ± 0.21 ^i^	0.00 ± 0.00

* Mean (n = 6) ± SE bearing different superscript(s) (a, b, c, d, e, f, g, h, i) in a column differ significantly (*p* < 0.05).

**Table 4 foods-13-02254-t004:** Estimation of MIC, MBC, and tolerance to ethanol extract of Ratanjot root extract powder (RRP).

Bacteria	RRP Ethanol Extract (mg/mL) *	Ciprofloxacin (µg/mL) *
MIC		
*Salmonella* Typhimurium ATCC 14028	25.00 ± 0.00 ^c^	0.125 ± 0.00 ^a^
*Escherichia coli* ATCC 25922	25.00 ± 0.00 ^c^	0.125 ± 0.00 ^a^
*Staphylococcus aureus* ATCC 25923	0.049 ± 0.00 ^a^	0.250 ± 0.00 ^b^
*Listeria monocytogenes* ATCC 19111	0.098 ± 0.00 ^b^	0.250 ± 0.00 ^b^
MBC		
*Salmonella* Typhimurium ATCC 14028	25.00 ± 0.00 ^c^	0.25 ± 0.00 ^a^
*Escherichia coli* ATCC 25922	25.00 ± 0.00 ^c^	0.25 ± 0.00 ^a^
*Staphylococcus aureus* ATCC 25923	0.098 ± 0.00 ^a^	0.50 ± 0.00 ^b^
*Listeria monocytogenes* ATCC 19111	0.195 ± 0.00 ^b^	0.50 ± 0.00 ^b^
Tolerance		
*Salmonella* Typhimurium ATCC 14028	1.00 ± 0.00 ^a^	2.00 ± 0.00
*Escherichia coli* ATCC 25922	1.00 ± 0.00 ^a^	2.00 ± 0.00
*Staphylococcus aureus* ATCC 25923	2.00 ± 0.00 ^b^	2.00 ± 0.00
*Listeria monocytogenes* ATCC 19111	2.00 ± 0.00 ^b^	2.00 ± 0.00

* Mean (n = 6) ± SE bearing different superscript(s) (a, b, c) in a column differ significantly (*p*< 0.05). MIC, minimum inhibitory concentration; MBC, minimum bactericidal concentration.

**Table 5 foods-13-02254-t005:** Effect of Ratanjot root extract powder (RRP) on sensory attributes of aerobically packaged chicken meatballs during refrigerated storage (4 ± 1 °C).

Sample	Refrigerated Storage Days *				
	Day 0	Day 5	Day 10	Day 15	Day 20
Color and Appearance					
Control	6.39 ± 0.07 ^aA^	6.28 ± 0.04 ^aB^	5.86 ± 0.06 ^aC^	5.44 ± 0.12 ^aD^	5.27 ± 0.14 ^aE^
T1	7.41 ± 0.05 ^bA^	7.33 ± 0.09 ^bB^	7.16 ± 0.05 ^bC^	6.94 ± 0.09 ^bD^	6.38 ± 0.21 ^bE^
T2	7.42 ± 0.04 ^bA^	7.37 ± 0.04 ^cB^	7. 24 ± 0.11 ^cC^	7.01 ± 0.05 ^cD^	6.63 ± 0.13 ^cE^
Flavor					
Control	6.37 ± 0.09 ^aA^	6.30 ± 0.13 ^aB^	5.86 ± 0.06 ^aC^	4.98 ± 0.10 ^aD^	4.04 ± 0.06 ^aE^
T1	6.52 ± 0.04 ^bA^	6.50 ± 0.20 ^bA^	6.26 ± 0.02 ^bC^	6.07 ± 0.17 ^bD^	5.63 ± 0.03 ^bE^
T2	6.76 ± 0.06 ^cA^	6.74 ± 0.21 ^cA^	6.33 ± 0.04 ^cB^	6.19 ± 0.11 ^cC^	5.81 ± 0.04 ^cD^
Texture and Tenderness					
Control	6.76 ± 0.18 ^aA^	6.68 ± 0.13 ^aB^	6.21 ± 0.07 ^aC^	5.69 ± 0.05 ^aD^	5.46 ± 0.21 ^aE^
T1	6.82 ± 0.05 ^bA^	6.80 ± 0.27 ^bA^	6.64 ± 0.06 ^bB^	6.23 ± 0.11 ^bC^	6.09 ± 0.18 ^bD^
T2	6.84 ± 0.06 ^bA^	6.82 ± 0.12 ^bA^	6.73 ± 0.09 ^cB^	6.44 ± 0.04 ^cC^	6.15 ± 0.21 ^cD^
Juiciness					
Control	6.25 ± 0.10 ^aA^	6.21 ± 0.17 ^aB^	5.69 ± 0.38 ^aC^	5.20 ± 0.07 ^aD^	5.04 ± 0.04 ^aE^
T1	6.28 ± 0.12 ^bA^	6.26 ± 0.15 ^abA^	5.86 ± 0.24 ^bB^	5.41 ± 0.10 ^bC^	5.16 ± 0.11 ^bD^
T2	6.30 ± 0.17 ^bA^	6.29 ± 0.04 ^bA^	5.98 ± 0.21 ^cB^	5.67 ± 0.10 ^cC^	5.23 ± 0.10 ^cD^
Overall Acceptability					
Control	6.84 ± 0.09 ^aA^	6.80 ± 0.04 ^aB^	5.68 ± 0.21 ^aC^	4.98 ± 0.38 ^aD^	4.02 ± 0.50 ^aE^
T1	6.89 ± 0.04 ^bA^	6.86 ± 0.08 ^bA^	6.04 ± 0.18 ^bB^	5.65 ± 0.17 ^bC^	5.06 ± 0.26 ^bD^
T2	6.92 ± 0.07 ^bA^	6.91 ± 0.10 ^cA^	6.17 ± 0.23 ^cB^	5.84 ± 0.15 ^cC^	5.13 ± 0.14 ^cD^

* Mean (n = 6) ± SE bearing different superscript(s) in a column (a, b, c) and in a row (A, B, C, D, E) differ significantly (*p* < 0.05). T1 contains 10 mg/g RRP and T2 contains 25 mg/g RRP.

## Data Availability

The original contributions presented in the study are included in the article/Appendix A, further inquiries can be directed to the corresponding authors.

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
