# Peer review of "Ratanjot (Alkanna tinctoria L.) Root Extract, Rich in Antioxidants, Exhibits Strong Antimicrobial Activity against Foodborne Pathogens and Is a Potential Food Preservative"

_foods, 2024, doi:10.3390/foods13142254_

Round 1
Reviewer 1 Report
Comments and Suggestions for Authors
This research article studied on the potential of the extract as a food preservative, however, it still needs clarification and improvement in the manuscript.
1. Abstract
Line 33 = Please check the extraction method – What does “air-dried” mean?
Line 37 = Please revise an appropriate term for GC-MS.
2. Introduction
· Line 58 = Add the references for each drawback and suggest pointing out the advantages of natural antioxidants or antimicrobials over synthetic ones.
· Line 68 -71 = These statements are misleading because not all plant extracts and essential oils are considered as GRAS according to the FDA.
· Line 90 -91 = Please add the toxicity profile information and what particular compounds are responsible for food colorant and flavoring agents.
3. Methodology
Line 105 -106 = The information about the chemical list and media needs to be provided.
Line 121 = The botanist’s institution has to be mentioned.
Line 155 = Please describe further the range of extract concentration. In addition, as the authors mentioned "the volume was made ten times" was not cleared.
Line 161 = As the authors tested various concentrations of the extracts, how did the authors determine the TPC value? Which concentration was chosen for reporting the data? As the concentration used, it may affect to the TPC value. In the result section, the authors report the single value of TPC. Please clarify it.
Line 168 = Please mention the range of concentrations tested.
Line 197 = Please describe the composition of the stationary phase used and the size of the column.
Line 198 = The data analysis should be described clearly by mentioning the method for determining the relative abundance.
Line 207 -208 = Please indicate the guideline used for the antimicrobial assay.
Line 211 = Please mention the number of colonies tested (cfu).
Line 219, 299, 465, 522, 558 = All species names should be written correctly.
Line 282 = Please add the ethical consideration document no. and information of the ethical review committee.
Line 301-303 = The sensory test by the panelist should be described clearly by mentioning all parameters used and the sample size. Also, please indicate the ethical clearance documents for performing this test. All methods must be described clearly.
4. Result and discussion
Line 337 = Please point out the guideline used by the authors judging the RRP extract as the highly valued colorant.
In Table 2, there are missing assignment of significant differences for b* parameter.
Line 400 – 401 = Please describe more detail regarding to the relation between DPPH assay and phenolic content.
Line 424 – 425 = As the authors did two kinds of RSA assays, please elaborate more on the implication of these two methods corresponding to the results obtained.
Figure 3, the structure of di-isooctylpthalate is incorrect. Please revise it.
Line 533 = Please indicate the possible reasons why RRP lost its antibacterial activity at pH 8.
Line 542 = "UD" annotation in Fig 5, should be indicated in the figure caption.
Line 580 = Why is triton-X used as the control to cause hemolysis? Consider using the synthetic antimicrobial as the authors highlighted its drawbacks compared to the natural one. Also, in this test, there is no control (without treatment). Please add the control to it, as it is very important for understanding the effect with and without treatment.
Line 607 = Please indicate the year of the guideline. All methodologies used, such as TBARS, TVC, psychrophilic counts, must be described clearly in the methodology section.
Table 5 = There is a lack of information on how the authors perform the sensory test. How did the authors test it? Please describe clearly the method for measuring the sensory attributes of the chicken meatball product and why the authors selected 4 ͦC for storage conditions.
Line 633 = What the implication of each value means in the table should be described. Also, in this section, it was suggested that the authors should provide the threshold of day storage with no significant change compared to the control.
5. Conclusion
Line 649 = The authors should mention the concentration tested as being non-toxic by the RBCs hemolytic assay.
Comments on the Quality of English Language
The authors should revise some typo errors.
Author Response
See attached response to the comments for Reviewer 1

Reviewer 2 Report
Comments and Suggestions for Authors
General comments
This manuscript evaluates the antioxidant and antimicrobial activity of Ratanjot (Alkanna tinctoria L.) root extract and its potential used as food preservative. The study is of interest to the field of food preservation. The experimental work is in general performed well providing new information. However, some clarifications about the methodology are necessary.
Specific comments
Abstract
Removed lines 37-39 since information about the color is not relevant.
Materials and methods
Add concentration of BHT and BHA used for controls of antioxidant activity.
A control system of ethanol is missing regarding the antimicrobial activity assays. Include this information.
Why swabbed instead of inoculation of the whole agar was used in the diffusion assay?
Why was the end point not determined by the change in absorbance or visual turbidity instead of using a colorant for MIC evaluation?(section 2.5.3)
Why did the authors used an inoculum from the log phase instead of the early stationary phase? It is well known that cells from the log phase are more sensitive to stress factors
Add more info about meatball formulation (section 2.8)
Hedonic scale must be used with consumers. Explain this fact.
Results and discussion
Since extract exhibited acidic pH. it is a must to include a control system at the pH the extracts.
In Figure 6, panel B is not necessary.
Comments on the Quality of English Language
No comments
Author Response
See attached responses to the comments of Reviewer 2
